# Grayscale Image Display Based on Nano-Polarizer Arrays

**DOI:** 10.3390/mi13111956

**Published:** 2022-11-11

**Authors:** Xinxin Pu, Xueping Sun, Shaobo Ge, Jin Cheng, Shun Zhou, Weiguo Liu

**Affiliations:** Shaanxi Province Key Laboratory of Thin Films Technology and Optical Test, School of Optoelectronic Engineering, Xi’an Technological University, Xi’an 710032, China

**Keywords:** nano-polarizer, metasurface, polarization state, grayscale display

## Abstract

Optical metasurfaces have shown unprecedented capabilities to control the two-dimensional distributions of phase, polarization, and intensity profiles of optical waves. Here, a TiO_2_ nanostructure functioning as a nano-polarizer was optimized considering that an anisotropic nanostructure is sensitive to the polarization states of incident light. We demonstrate two metasurfaces consisting of nano-polarizer arrays featured with different orientations, which can continuously manipulate the intensity distribution of the output light cell by cell according to Malus law and clearly display the detailed information of the target image. These metasurfaces have potential application in ultracompact displays, high-density optical information storage, and many other related polarization optics fields.

## 1. Introduction

Grayscale images or binary images play an important role in our daily life. For example, binary grayscale display is usually used in LED displays and grayscale display is often used to transfer textual information. In the processing of micro-nano structures, gray-tone lithography is employed to produce continuous profile microstructures, which requires a grayscale image with high-density storage acting as a photomask to transfer the design layout to the photoresist. With the development of display technology, the grayscale image needs to meet requirements such as high resolution, ultracompactness, feasible fabrication, and high storing density, among others [1,2].

An inhomogeneous distribution of polarization may lead to new effects or phenomena that can expand the functionality and enhance the capability of optical systems. For instance, the radially polarized field, as a typical vector field, can be focused into a far-field focal spot of 0.17l, beyond the diffraction limit of 0.261 [3]. In particular, the azimuthally polarized vector field (another typical vector field) carrying a vortex phase can realize a sharper far-field focal spot of 0.15l [4]. Although many methods to generate vectorial optical fields have been explored [5,6,7], such as fiber lasers [8], vector light field generators [9], light-aligned liquid crystals [10], and digital micromirror devices [11], the application of vector light fields is still limited owing to the complexity of optical systems and the inability to control the polarization state of light at the pixel level [12].

The optical metasurface, consisting of a series of arrays of subwavelength micro-nano structures, provides a way to flexibly modulate the optical properties such as amplitude and polarization of electromagnetic wave, and is a promising candidate to alleviate the above problems [13]. Recently, a lot of work has been carried out taking advantage of metasurfaces, which demonstrated various new functions and applications, such as broadband achromatic metalenses [14,15,16], dynamic holography [17,18,19,20,21,22,23,24], optically resonant dielectric and semiconductor metasurfaces [25,26,27,28,29], among others. Of particular interest is the anisotropic unitary nanostructure, which is sensitive to the polarization state of the incident light. Meanwhile, the anisotropic nanostructure has different amplitude and phase modulation to incident light with different polarization states, which makes a metasurface capable of producing vector beams with a specific polarization distribution and displaying grayscale images at the subwavelength pixel level. Based on a reflective metal insulator metal (MIM) metasurface, Yue et al. proposed an approach to encode a high-resolution grayscale image into the polarization profile and decode the image information after passing through a linear polarizer [30]. They further demonstrated a high-efficiency transmissive dielectric metasurface for simultaneously encoding color and intensity information into the wavelength-dependent polarization profile [11]. However, an additional linear polarizer is required to decode the image information in their research. Dai et al. proposed a nano-polarizer array to decode a high-fidelity grayscale image with resolution as high as 84667 dpi (dots per inch) [31]. Subsequently, benefiting from the orientation degeneracy implied in Malus law, they proposed many metasurfaces with different functions [1,31,32,33]. In a dual-channel display field, they proposed a Malus-assisted metasurface [1] with a single-sized nanostructure and a multiplexed anticounterfeiting metasurface [32]; the former can store two different gray images without crosstalk, while the latter can record a continuous grayscale image multiplexed with a watermarked anticounterfeiting pattern. Following this, they designed an ultrathin and single-sized metasurface, which can generate a near-field grayscale pattern and project an independent far-field holographic image [33]. Some metasurfaces for three-channel image display have also been proposed by combining intensity modulation with phase manipulation based on both Pancharatnam−Berry (PB) and propagation phases [34,35,36]. The two- or three-channel metasurfaces can empower advanced applications in many fields such as information multiplexing and encryption [37], multichannel image display [38,39,40], and optical anticounterfeiting [41]. Among the above metasurfaces, some of them are realized by nanostructures acting as wave-plates and an additional analyzer is required to decode grayscale image information in this way. The others are realized by reflective nanostructures, which are limited by reflective operation that is not compatible with most optical systems that operate in the transmission mode.

In this paper, we propose a transmissive nano-polarizer, composed of a TiO_2_ nanobrick sitting on an SiO_2_ dielectric substrate. By elaborately designing each of the nano-polarizer orientations, we can achieve the pixel-scale polarization distribution modulation of the output light. Based on Malus law, precise and continuous intensity distributions of the transmitted light can also be attained on the surface of the metasurface. We achieve the high-resolution and continuous grayscale image storage and display at the pixel size of 510 × 510 nm.

## 2. Unit Cell Design

The unit cell designed here is a TiO_2_ nanobrick sitting on an SiO_2_ substrate. As shown in Figure 1, the length, width, and height of the TiO_2_ nanobrick are denoted as L, W and H, respectively. The period of the unit cell both in x and y directions is P and its long and short axes are along the x and y directions, respectively. When the length and width of the nanobrick are different, the electromagnetic responses to x- and y-linearly polarized light incidence become different. For a linear polarizer, the intensity of the output light is equal to the intensity of the incident light when the polarization orientation of the incident light is the same as the transmittance axis of the polarizer. On the other hand, when the polarization direction of the incident light is orthogonal to the transmission axis of the polarizer, the intensity of the output light is zero. More generally, when the polarization orientation of the incident light with intensity *I*_0_ and the direction of the transmitting axis of the polarizer are at an angle *θ*, the intensity of the output light is $I=I_{0}{cos}^{2}\left(\theta \right)$, namely, the intensity of the output light satisfies Malus law.

In order to realize a TiO_2_ nanobrick satisfying Malus law, i.e., a nano-polarizer, we use the finite-difference time domain method to investigate the transmittance characteristics of different TiO_2_ nanobrick structure parameters and different unit cell periods. These nanobricks with transmittance greater than 95% for x- linearly polarized light and less than 5% for y- linearly polarized light are screened out firstly. However, not all of them satisfy Malus law (a counterexample will be given later), and the nanobricks satisfying the above requirements need to be further verified under different polarized light incidence. After optimizing the structural parameters of these nanobricks, we obtained a nano-polarizer that satisfies Malus law. The structural parameters of this nano-polarizer have a length of 80 nm, width of 210 nm, height of 400 nm and period of 510 nm. For the optimized nano-polarizer upon normal x- and y- linearly polarized light incidence, the transmittances $T_{x}$ and $T_{y}$ in a broad wavelength range between 400 nm and 800 nm are shown in Figure 2a. From Figure 2a, it was found that the transmittance $T_{x}$ maintains a high value over 95% and $T_{y}$ reaches nearly zero with a wavelength range from 540 nm to 580 nm. Especially, the transmittances of $T_{x}$ = 96.2% and $T_{y}$ = 0.3% are obtained at the design wavelength *λ* = 568 nm.

In order to show that the optimized nanobrick can act as a polarizer, we rotated the nanobrick along the z-axis and set its rotation angle as *θ*. The transmittance distribution when the x-linearly polarized light passed through the rotated nanobrick is shown in Figure 2b, and it can be found that the transmittance curve is well consistent with Malus law. Comparatively, we also provide a nanobrick with transmittance satisfying $T_{x}$ > 95% and $T_{y}$ < 5%, but it cannot meet Malus law. The corresponding structure parameters are *L* = 56.5 nm, *W* = 325 nm, *H* = 370 nm, and *P* = 370 nm. The transmittances $T_{x}$ and $T_{y}$ in the visible wavelength under the irradiation of x- and y- linearly polarized light are shown in Figure 3a. It can be seen that, at the wavelength of 568 nm, $T_{x}$ = 98.9%, and $T_{y}$ = 0.4%, it seems that, at 568 nm, the transmittance spectrum has a curve Fano resonance. When the light is incident on the designed unit cell, the dipole mode (i.e., bright mode) can usually be excited directly. The weak dark mode (narrow line mode with very small net dipole moment) cannot be excited directly. However, it can be excited by means of near-field coupling of the bright modes. The dark mode is excited by the bright mode and then reacts to the bright mode by near-field coupling, so we consider that the spectrum that appears to be a Fano resonance curve is the result of the Fano resonance. The transmittance distribution when x-polarized light passed through the rotated nanobrick is shown in Figure 3b. It is obvious that the transmission curve of this nanobrick does not conform to Malus law, because the dark mode of the adjacent unit cell that does not satisfy Malus law will be excited by the bright mode through the near coupling. As the distance between adjacent unit cells decreases with the increasing rotation angle, the coupling becomes stronger. It is the coupling that excites the strong dark mode, which leads to the structure being inconsistent with Malus law. In short, the reason for this phenomenon was the strong coupling effect between adjacent unit cells. We should avoid this phenomenon in the design, because the near-field coupling effect is too strong, so the design of the unit cell cannot be modulated according to the requirements of the light.

To illustrate that the designed nano-polarizer is scalable by the wavelength, we modified the length, the width, the height, and the period of the designed nano-polarizer by wavelength as $L=\frac{\lambda }{\lambda_{0}}L_{0}$, $W=\frac{\lambda }{\lambda_{0}}W_{0}$, $H=\frac{\lambda }{\lambda_{0}}H_{0}$, and $P=\frac{\lambda }{\lambda_{0}}P_{0}$, respectively, and here we set *λ* = 632.8 nm. $L_{0}$, $W_{0}$, $H_{0}$ and $P_{0}$ are the parameters of our designed nano-polarizer. After modelling and simulating the scaled nano-polarizer, we obtain the transmittance with the rotation angle of the nanobrick, as presented in Figure 4. In our simulation, the boundary conditions are periodic boundary conditions in the x-and y-directions and perfectly matched layer boundary conditions in the z-direction. As can be seen in Figure 4, the scaled nanobrick still acts as a polarizer at the design wavelength *λ*, and this verifies that our designed polarizer is scalable by wavelength.

## 3. Grayscale Image Design

Figure 5 presents a schematic illustration of the continuous grayscale image display based on the nano-polarizer arrays. In this section, we use the x-linearly polarized light as the incident. By choosing the short axis (direction L) of the nano-polarizer along the x-direction, the transmitted light would get its maximum $I_{max}$ that the transmitted light nearly tends to 0 when the long axis (direction W) is along the x-direction. To obtain an arbitrary light intensity *I* between 0 and $I_{max}$, Malus law is applied to achieve the precise intensity modulation by rotating the nano-polarizer. As the minimum transmittance is only 0.3%, we can ignore this influence and consider it as a polarizer. Rotating the nano-polarizer by an angle of $\theta =\arccos\left(\sqrt{\frac{I}{I_{\max}}}\right)$ helps us to achieve accurate intensity I. When the gray level of each pixel in the grayscale image meets the requirements, the output light will display the grayscale image of the input image.

In our daily life, binary images can be seen everywhere, such as books, banners, LED display, and so on. Therefore, we first display a binary grayscale image of “西安工业大学”, which has 151 × 51 pixels, as shown in Figure 6a. The orientation angle of the nano-polarizer is 0 or 90°. We place a nano-polarizer rotated by 90° for pixels with gray level 0 and a nano-polarizer without any rotation for pixels with gray level 255. Figure 6b shows the arrangement of the nano-polarizer array. When a beam of x-linearly polarized light illuminates this designed nano-polarizer array, the output near-field light intensity distribution is shown in Figure 6c. It can be seen from Figure 6a,c that the distribution of the output light intensity is consistent with the target image, which shows the desired text.

Next, we displayed the university logo of Xi’an Technological University. The original grayscale image is shown in Figure 7a, which has 151 × 151 pixels and a gray level of 255. The orientation angles of the nano-polarizer arrays vary from 0° to 90°. As the transmitted intensity can be modulated by the nano-polarizer pixel-by-pixel, we can place a suitably rotated nano-polarizer at each pixel to obtain the nano-polarizer array arrangement, as shown in Figure 7b. The corresponding output near-field light intensity distribution is shown in Figure 7c, which clearly shows the details of the university logo such as the numbers, words, and Chinese characters. From the results of the simulation, we can modulate the intensity of the transmitted near-field beam at the pixel level, which also shows that we can use nano-polarizer to achieve high-performance image display.

## 4. Conclusions

We proposed a nano-polarizer that satisfies Malus law, which is scalable by the design wavelength. This nano-polarizer provides a way to control the polarization orientation of polarized light, which can achieve polarization distribution modulation at the subwavelength scale. Therefore, the light intensity of near-field transmitted light can be manipulated by the polarization distribution of transmitted light under the government of Malus law. Based on the proposed nano-polarizer, a target binary image and a grayscale image were clearly reproduced and realized a precise and continuous grayscale modulation of these. We believe that the nano-polarizer has great potential for applications in polarization optics, which can miniaturize and integrate polarization optical systems and many other related applications such as high-density optical storage and multifunctional metasurfaces.

## Figures and Tables

**Figure 1 micromachines-13-01956-f001:**
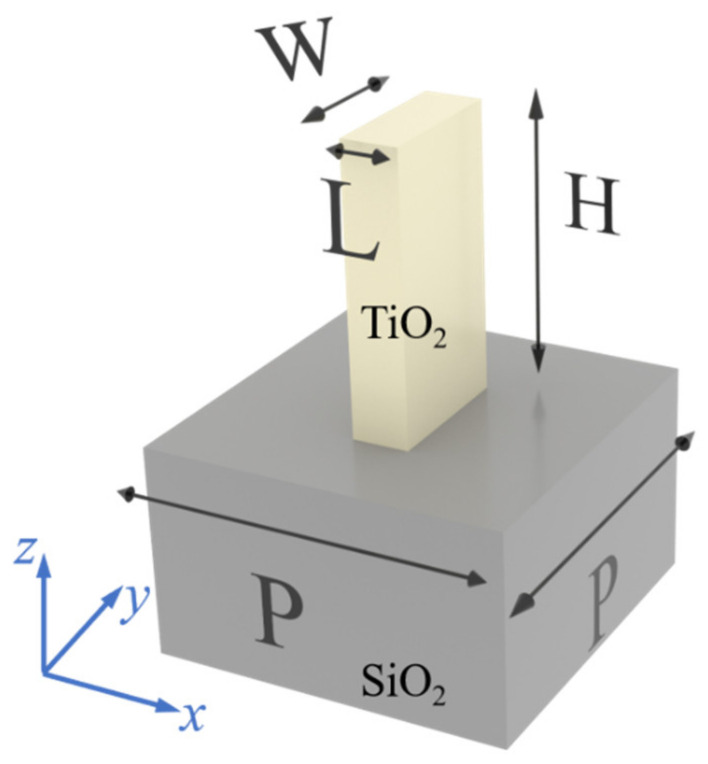
A unit cell with a TiO_2_ nanostructure standing on an SiO_2_ substrate.

**Figure 2 micromachines-13-01956-f002:**
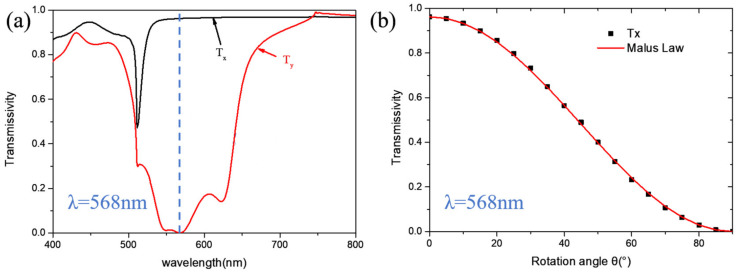
The structural parameters are L = 80 nm, W = 210 nm, H = 400 nm, and P = 510 nm. (**a**) The distribution of transmittances $T_{x}$ and $T_{y}$ in the visible light. (**b**) The transmittance distribution of x-linearly polarized light through the rotated unit cell.

**Figure 3 micromachines-13-01956-f003:**
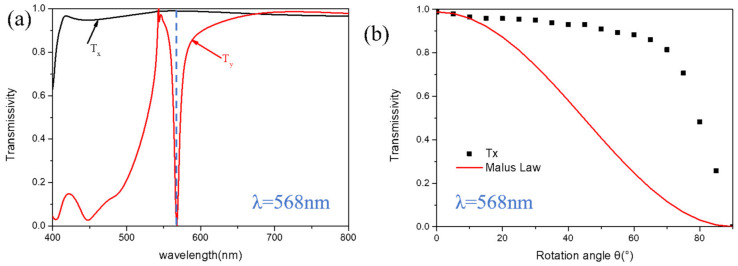
The structural parameters are L = 56.5 nm, W = 325 nm, H = 370 nm, and P = 370 nm. (**a**) The distribution of transmittances $T_{x}$ and $T_{y}$ in the visible light region. (**b**) The transmittance distribution of x- linearly polarized light through the rotated unit cell.

**Figure 4 micromachines-13-01956-f004:**
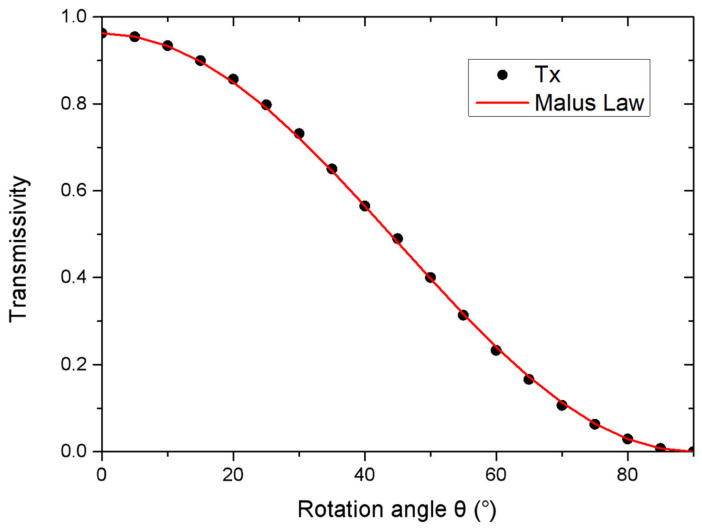
The curve of transmittance changing with the rotation angle of the nano structure.

**Figure 5 micromachines-13-01956-f005:**
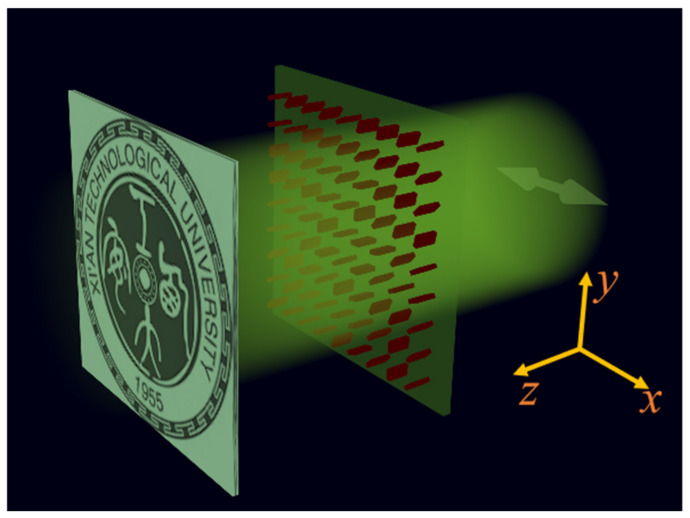
The schematic illustration of the continuous grayscale image display based on the nano-polarizer arrays.

**Figure 6 micromachines-13-01956-f006:**
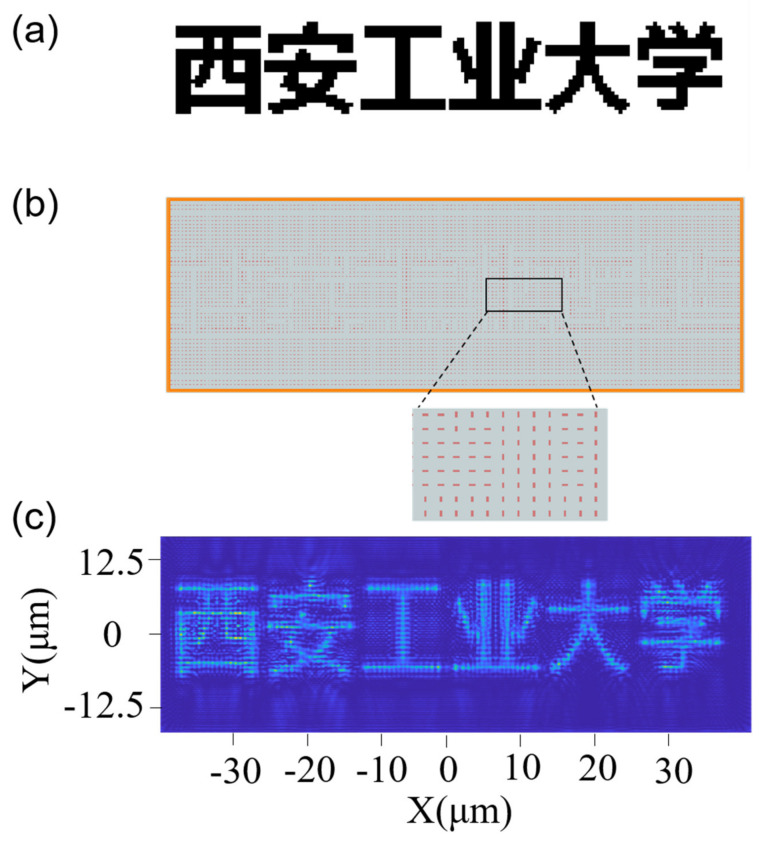
(**a**) A binary grayscale image. (**b**) The arrangement of the nano-Polarizer array. (**c**) The near-field light intensity distribution of the output light.

**Figure 7 micromachines-13-01956-f007:**
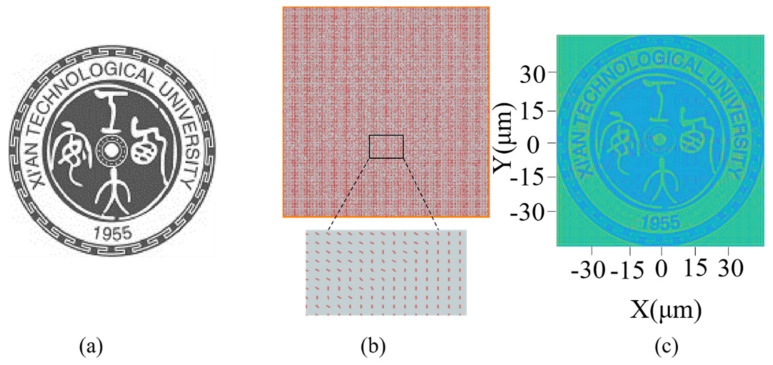
(**a**) The target image. (**b**) The arrangement of the nano-Polarizer array. (**c**) The near-field light intensity distribution of the output light.

## Data Availability

Not applicable.

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
