# Peer review of "Grayscale Image Display Based on Nano-Polarizer Arrays"

_micromachines, 2022, doi:10.3390/mi13111956_

Round 1
Reviewer 1 Report
In this work, Pu et al have computationaly demonstrated a polarization-selective metasurface in the visible wavelength range. The metasurface is made with a TiO2 nano fin sitting on top of a SiO2 substrate (Fig. 1), which has a high transmision for light whose polarization is perpendicular to the longer axis of the fin, but a low transmission for parrallel polarization (Fig. 2a). By orienting the nanofin differnetly at different locations, they were able to generate two target transmission profiles (Fig. 6c & Fig. 7c).
The paper includes a detailed literature review of the related work, and presents the simulation results clearly.
I have the following major and minor suggestions which authors may consider
Major concerns:
1. the title claims the demonstration of a grayscale image display, which is misleading because the paper did not show a display (with emission layer, electric drive, cavity, hole/electron transport layer etc). Instead, the paper shows a polarization-selective metasurface, which can produce a target transmission profile for a polarized light, but cannot be actively tuned (at least not presented in the paper).
2. the simulation results (including reproducing two target images) are demonstrated in the near field only, the far field analysis is not presented, which limits many of the potential applications including in the aforementioned display technologies. Authors may want to reconsider the impact (i.e. potential applications) of near field modulation, or include far field analysis.
3. Authors credit Malus law when explaining how does the demonstrated metasurface work. However, Fig. 3(b) clearly shows a violation of the Marlus law. The violation is attributed to the coupling effect between adjacent meta-atoms, but this claim does not have any reference nor simulation results to support it. Without a clear explaination of the violation case, the light modulation mechanism is not well understood.
4. The physics behind the simulated result is unknown. Why the demonstrated metasurface transmits a certain polarization but not the other is not well explained. For example in Fig.2 (a), even though the transmittance for x-polarized light is >95%, the transmission of y-polarized light is almost 0, where does the y-polarized light go? Reflected? or absorbed? Is the selectivity due to traveling phase difference? Or plasmon resonance?
Minor comments:
1. p2 line 67, "PB" should be spelled out since here is the first time the term appears in this paper
2. p4 line 132 (caption of Fig 3). The precision of different geometric dimensions given is inconsistent. L is given with 0.1nm precision, while others are given with 1nm precision. How sensitive is the transmission to each geometric features? If the tolerance is below 10nm, can that be realized with the state-of-art lithography technique?
3. p4 line 146 (caption of Fig 4). "micro-nano structure" is a confusing term, authors probably meant nano structure. This caption is too short and is not clear. What is this figure trying to show? What makes this figure (obeys Marlus law) different from figure 3b (violates Marlus law)?
4. p5 line 151~152. The sentense does not make sense to me, authors want to check the grammer errors.
5. p5 line 155. The statement of treating the metasurface as an ideal polarizer is not objective and does not provide any technical information.
6. also p5 line 156. The font or line spacing is off for this line, which makes it look wider than other lines.
7. p5 line 162. This paragraph (describing background and summary) can be put at the begining of this section.
8. p6 Fig 6. (b) why are some pixels missing pattern? (c) need to add a scale bar for the simulated pattern. May want to comment on why there are residue field around the text.
9. p7 line 17~18. The sentense claims that the demonstrated metasurface "provides a way to control the poalrization orientation of polarized light", which is not accurate. The metasurface has different transmission depending on the poarlization oreintation of the illluminating polarized light, but the paper did not demonstrate the modulation of the polarization.
Author Response
Please see the attachment,thank you very much.

Reviewer 2 Report
In this manuscript, authors proposed a TiO2-based nanostructure polarizer for image display. The working principle was elaborated based on Malus’s Law. However, some shortcomings still exist in this manuscript. Thus, I would recommend this work for publication in the journal of Micromachines after addressing the following the remarks:
1. The introduction section may be expanded to include more details on existing work reported in literature, highlighting gaps to establish grounds for the proposed work to demonstrate its significance.
2. In Page 3, Why do you choose 568nm? It's just an optimal result at the corresponding size? Or is the target value for optimization 568nm?
3. In page 4, authors mentioned that the phenomenon of breaking the Malus Law was the strong coupling effect between adjacent unit cells. Authors should give (show) the direct evidence to support this reason.
4. In page 4, why is the operating wavelength 632.8nm when analyzing the scaled size structure? I think it's just a very strange wavelength, what is the relation between the wavelength of 568nm mentioned earlier? Or is it also another arbitrary structure that determines a wavelength?
5. what does λ0 mean in page 4 line 137?
6. The authors should give the simulation process and method of imaging in detail.
Author Response
"Please see the attachment, thank you very much.

Reviewer 3 Report
Please find my comments in the attached PDF file.
Best Regards.

Author Response
Please see the attachment, thank you very much.

Round 2
Reviewer 1 Report
Thank the authors for detailed response. However, I do not believe the manuscript has been sufficiently improved. The changes made from the last version are very limited.
I recommend changing the title and adding the response to my major concern 3-4 to the main body.
On my minor comments:
2. Still, I would like authors to comment on how sensitive is the transmission to the feature size of the meta atoms.
3. I suggest authors to directly use the term "nano brick" (without hyphen)
5. I suggest removing the descriptive term "ideal" - there can be many criteria for the term which are not illustrated in the paper, i.e. is the selectivity 100%? is absorption 0%? etc
8. The authors may have misunderstood my question by claiming a 1:1 scale that can show nanometer features on the screen. A scale bar of micron scale will make more sense to me. Also, please include the rest part of the response in the main body to avoid confusion.
Author Response
Please see the attachment, thank you for your valuable comments.

Reviewer 3 Report
Please consider my comments in the attached pdf file.
Best Regards.

Author Response

(The authors gave the same response as above.)

Round 3
Reviewer 3 Report
The manuscript has improved and the revised version of the paper may be published as it is.
Best Regards.